# Role of Estrogens in Menstrual Migraine

**DOI:** 10.3390/cells11081355

**Published:** 2022-04-15

**Authors:** Rossella E. Nappi, Lara Tiranini, Simona Sacco, Eleonora De Matteis, Roberto De Icco, Cristina Tassorelli

**Affiliations:** 1Research Center for Reproductive Medicine, Gynecological Endocrinology and Menopause, IRCCS San Matteo Foundation, 27100 Pavia, Italy; 2Department of Clinical, Surgical, Diagnostic and Pediatric Sciences, School of Medicine, University of Pavia, 27100 Pavia, Italy; lara.tiranini01@universitadipavia.it; 3Neuroscience Section, Department of Applied Clinical Sciences and Biotechnology, School of Medicine, University of L’Aquila, 67100 L’Aquila, Italy; simona.sacco@univaq.it (S.S.); eleonoradematteis@fastwebnet.it (E.D.M.); 4Headache Science and Neurorehabilitation Center, National Neurological Institute C. Mondino Foundation, 27100 Pavia, Italy; rob.deicco@gmail.com (R.D.I.); cristina.tassorelli@unipv.it (C.T.); 5Department of Brain and Behavioral Sciences, School of Medicine, University of Pavia, 27100 Pavia, Italy

**Keywords:** reproductive hormones, calcitonin gene-related peptide, contraception, hormone replacement therapy, ethinylestradiol, efficacy, gender, estradiol, progesterone, regimen

## Abstract

Migraine is a major neurological disorder affecting one in nine adults worldwide with a significant impact on health care and socioeconomic systems. Migraine is more prevalent in women than in men, with 17% of all women meeting the diagnostic criteria for migraine. In women, the frequency of migraine attacks shows variations over the menstrual cycle and pregnancy, and the use of combined hormonal contraception (CHC) or hormone replacement therapy (HRT) can unveil or modify migraine disease. In the general population, 18–25% of female migraineurs display a menstrual association of their headache. Here we present an overview on the evidence supporting the role of reproductive hormones, in particular estrogens, in the pathophysiology of migraine. We also analyze the efficacy and safety of prescribing exogenous estrogens as a potential treatment for menstrual-related migraine. Finally, we point to controversial issues and future research areas in the field of reproductive hormones and migraine.

## 1. Introduction

Migraine is a major neurological disorder affecting one in nine adults worldwide with a significant impact on health care and socioeconomic systems [1]. The most frequent subtype is migraine without aura, whereas one in four migraineurs experience an aura, namely, a cluster of transient positive or negative neurological symptoms before headache onset each lasting 20–30 min [1]. The prevalence of migraine during childhood is similar in boys and girls, but after puberty, it rises differently in sexes becoming two to three times more prevalent in women than in men, with 17% of all women meeting the diagnostic criteria for migraine [2,3]. In the female lifespan, the clinical pattern of migraine is linked to reproductive milestones with an increase around puberty, a peak during fertile age, and a decline after menopause [4,5]. Furthermore, the frequency of migraine attacks shows variations over the menstrual cycle and pregnancy, and the use of combined hormonal contraception (CHC) or hormone replacement therapy (HRT) can unveil or modify migraine disease [6,7,8,9,10]. In the general population, 18–25% of female migraineurs display a menstrual association of their headache [11]. Notably, in the Appendix of the International Classification of Headache Disorders 3 (ICHD-3), the diagnostic criteria identify pure menstrual migraine (PMM) with or without aura occurring exclusively in the perimenstrual period (from −2 to +3 days), and menstrual-related migraine (MRM) with or without aura occurring both perimenstrually and at other times of the menstrual cycle [12].

In this paper, we use the terminology menstrual migraine (MM) to indicate both PMM and MRM when studies were conducted without referring to distinct diagnostic criteria [12]. Our aim is to present an overview of the evidence supporting the role of reproductive hormones, in particular estrogens, in the pathophysiology of migraine [13]. We also analyze the efficacy and safety of prescribing exogenous estrogens as a potential treatment for MM. Finally, we point to controversial issues and future research areas in the field of reproductive hormones and migraine.

## 2. Role of Estrogens and Estrogen Receptors in the Pathophysiology of Migraine

The first evidence of the association between estrogens and migraine dates back to 1972, when Somerville demonstrated that the intramuscular injection of estradiol valerate (a pro-drug ester of 17 β-estradiol [E2]), but not of progesterone, shortly before menstruation delayed the onset of menstrual migraine in female subjects with migraine [14]. Following a period of priming with exogenous estrogens, a threshold of circulating E2 levels (45–50 pg/mL), below which migraines were especially triggered, was identified [14]. The existence of an estrogenic threshold was subsequently confirmed by studying postmenopausal women taking HRT following the intramuscular injection of E2 [15]. These findings implied that migraine attacks associated with menstruation could be due to an estrogen withdrawal effect [10,11,12,13,16]. Apart from the link between a drop in estrogen and migraine onset after prior estrogen exposure, nowadays it seems likely that physiologic estrogen fluctuations also play a role in migraine pathogenesis [11]. Indeed, a specific neuroendocrine vulnerability in female migraineurs was evident when the rate of estrogen decline in the late luteal phase of patients was compared with that of healthy controls [17]. However, no differences in the peak level of estrogen or mean daily levels during ovulatory cycles were demonstrated when female patients with migraine were compared with healthy controls [17].

### 2.1. Estrogens and Estrogen Receptors

Estrogens are lipophilic hormones derived from cholesterol and synthetized primarily in the granulosa cells of ovaries. They also derive from the aromatization of androgens in peripheral tissues and within the brain [18]. The most relevant endogenous estrogen is E2, the classical female sex steroid hormone. Estrogens reach the central nervous system (CNS) passively diffusing through the blood–brain barrier, but they are also locally synthetized from cholesterol or converted from aromatizable precursors by the brain enzyme aromatase, thus acting as neurosteroids [19]. Estrogens achieve physiological effects through the activation of various estrogen receptors (ERs), including three known forms: estrogen receptor-α (ERα), estrogen receptor-β (ERβ), and the more recently identified G protein-coupled estrogen receptor-1 (GPER/GPR30) [18].

ERα and ERβ are two classical nuclear receptors that mediate the effects of estrogens on gene expression. They belong to the nuclear receptor superfamily and are constituted by four domains, including a ligand-binding domain and a DNA-binding domain [20]. Inactivated ERs are mainly located in the cellular nucleus (nearly 95%), but in contrast to the original hypothesis that they are pure modulators of DNA transcription, the remaining 5% of ERs localize to the plasma membrane and to cytoplasmic organelles including the endoplasmic reticulum and mitochondria [21,22]. Thus, ERs also act through second messenger signaling such as MAPK (mitogen-activated protein kinase) and PKB (protein kinase B) pathways [23]. The binding between inactive ER and its ligand determines the activation of the receptor through a conformational change, leading to a dissociation from the chaperone (Hsp90), dimerization, and in the case of cytosolic ER, nuclear translocation. In the nucleus, the ER dimer acts as a sequence-specific DNA-binding protein that recruits co-regulators such as CREB (cAMP response element-binding protein) and SRC-1 (steroid receptor coactivator-1) and binds to the estrogen response element (ERE) in the promoter region of the targeted gene for the regulation of transcription [22,24,25,26]. The third ER, GPER, is a 7-transmembrane G-protein-coupled receptor that specifically binds to estrogens [27]. GPER is mainly localized in the cytoplasmic membrane [28], but it can also be in the nucleus [29]. GPER regulates a variety of cellular functions through the activation of intracellular signaling cascades such as ERK1/2 and PI3K/PKB [30], as well as through the modulation of gene expression [22]. Consequently, estrogens exert their biological effects in the CNS either by genomic or non-genomic cellular mechanisms [31]. Genomic transcriptional regulation mediated by ERs often takes hours to days to manifest. It is achieved through direct binding to nuclear ERs (classical genomic mechanism) but also through the activation of the membrane or cytoplasmic receptors. Subsequent secondary messenger signaling leads to the final molecules modulating gene transcription (non-classical genomic mechanism) [22,25,32]. Rapid, non-genomic estrogen-dependent signals also influence neurotransmission and cell function. Indeed, the binding of estrogen to its membrane or cytoplasmic receptor activates an intracellular signaling cascade that modifies enzymatic pathways, the conductance of ion channels, and neuronal excitability [18,33,34].

Animal studies have revealed that numerous brain areas involved in migraine pathophysiology and pain processing express ERs [35,36]. All three receptor subtypes have been identified in the hypothalamus, a critical region for migraine initiation [37], and in the cerebellum [38]. ERα is also expressed in the ventral striatum of the limbic system, pontine nuclei, and the periaqueductal grey region [38,39]. ERβ is mainly present in the hippocampus [38] and locus coeruleus [40]. GPER is found in the amygdala, hippocampus, pontine nuclei, and trigeminal nucleus caudalis [36,38]. Both ERα and ERβ are expressed in the cerebral cortex, suggesting that estrogens could modulate pain perception at the highest cognitive level and influence pain sensitivity through descending pathways [36]. These ERs have also been found in the smooth muscle cells and endothelial cells of cerebral arteries [38]. Finally, both the trigeminal ganglion (TG), a cluster of sensory neurons that innervate dura mater and cranial vasculature, and the dorsal horn of the spinal cord, the first relay point in the trigeminal transmission of painful stimuli from the periphery, express all three ERs [36,41].

Genetic studies have explored a possible role for ER variants in the pathogenesis of migraine. Genome-wide association analysis did not find a correlation between estrogen receptor-1 gene (ESR1) polymorphisms and an increased susceptibility to migraine [42]. Variants in the estrogen metabolism genes COMT, CYP1A1, and CYP19A1 were not associated with MM [43]. By contrast, single nucleotide polymorphisms in the SYNE1 gene (spectrin repeat-containing nuclear envelope 1), a gene adjacent to ESR1, were positively associated to MM [44]. Additionally, the neuropilin 1 gene (NRP1) encodes for a transmembrane protein correlated with MM [45]. The role of E2 as an epigenetic modulator in the methylation of genes involved in migraine pathophysiology including ESR1 was investigated without demonstrating a significant effect [46]. Future research assessing the effect of estrogen withdrawal on gene methylation may be of interest [11].

### 2.2. Estrogens and Neurotransmitter Systems

Estrogens can modulate the activity of several neurotransmitter systems involved in the pathophysiology of migraine and the pain network pathways [11]. The effect on these systems may vary according to estrogen concentrations, synergistic or antagonistic interactions with other neurosteroids, acute or chronic exposure, and the different expression of receptor subtypes in specific brain areas and nervous cells [25]. The serotonergic system is enhanced by estrogens and has a protective role toward migraine attacks [11]. Indeed, estrogens regulate the gene expression of serotonin (5-HT) receptors [47], promote 5-HT synthesis through increased levels of tryptophan hydroxylase [48], and decrease 5-HT degradation and reuptake through the modulation of monoamine oxidases (MAO-A, MAO-B) [49], and the 5-HT reuptake transporter (SERT) [50], respectively. Of note, E2 supplementation restored modulatory pain pathways involving 5-HT2 and 5-HT1 receptors in women with severe MM induced by the hormone-free interval of CHC, a good model to study the “perimenstrual window” of vulnerability to head pain [51]. That being so, a change in 5-HT tone accompanying estrogen withdrawal may be a trigger for MM [11]. Conversely, estrogens enhance the excitatory neurotransmission of glutamate, thus explaining the increased risk of migraine aura in high estrogen states, for instance during pregnancy or under exogenous hormone use [11]. Indeed, estrogens induce the formation of new spines and functional synapses in glutamatergic neurons [52] and also increase the transcription of genes encoding for glutamate receptors [25]. The gamma-aminobutyric acid (GABA) system, the major inhibitory neurotransmitter system in the CNS, is strongly modulated by estrogens that increase glutamic acid decarboxylase levels, GABA release, and the density of GABA receptors [53]. Both progesterone, a reproductive hormone mainly produced by the ovarian corpus luteum, and its metabolite allopregnanolone enhance GABAergic activity, thus promoting an antinociceptive action [54]. Another important neuroendocrine target of estrogens is the opioid system, which is highly relevant to several types of headache [55]. Estrogens modulate the opioid system through an increased synthesis of the opioid receptor ligand enkephalin [56], an enhanced binding affinity to opioid receptors [57], and an alteration of opioid receptor interaction with ion channels [25]. Of note, women with MRM displayed a transient premenstrual failure of central opioid tonus [58]. Indeed, during the late luteal phase, the low estrogen (and low progesterone) state was associated with a reduced activation of the opioid system, thus resulting in an enhanced susceptibility to pain [59]. Interestingly, when the sensitivity to pain stimuli in healthy women was assessed during different phases of the menstrual cycle, a reduced nociceptive threshold over the luteal phase, especially in those women reporting premenstrual complaints, was evident [60]. Finally, high estrogen levels promote the expression of other antinociceptive molecules, such as neuropeptide Y, prolactin, vasopressin, ghrelin, and galanin, which contribute to the complex influence of estrogens on migraine etiology [36,61].

### 2.3. Estrogens and Oxytocin

Novel trends point to the role of oxytocin (OT) in modulating the impact of estrogen withdrawal in MM [36]. OT is a neuropeptide produced in the hypothalamus and released either in the brain through neuronal projections or in the peripheral circulation through the posterior pituitary gland. OT has widespread effects within the CNS, regulates mood and behavior, and plays a role in pain suppression [62] including migraine prevention [63]. Indeed, hypothalamic regions that contribute to migraine initiation are also sources of OT, with projections throughout the brain, the spinal cord, and the posterior pituitary [64,65]. Moreover, OT receptors are expressed in migraine-related brain areas, as well as in the TG, trigeminal nucleus caudalis (TNC), and spinal cord [64,65], thus suggesting an influence on pain processing and migraine pathophysiology [36]. Interestingly, when estrogen levels are increased, OT is also high [36] and during the menstrual cycle fluctuations in the plasma levels of OT reflect those of estrogens [66]. The hypothalamic OT production is increased especially via ERβ [67], and estrogens upregulate the expression of OT receptors through genomic and non-genomic responses mediated by both ERα and ERβ in many brain areas, including the TG [36].

### 2.4. Estrogens and the Trigeminovascular System

The pathophysiology of migraine attacks is associated with the trigeminovascular nociceptive pathway because of activated cortical nociceptive projections, alterations in intracranial vasculature, or meningeal inflammation [36,68,69]. Indeed, neurons of TG process the peripheral nociceptive information, which, in turn, travels to the CNS via projections to TNC and the dorsal horn of the upper cervical spinal cord [70]. In here, second-order neurons further project to the subcortical and cortical pain regions [71]. Moreover, the pseudo-unipolar sensory neurons within the TG directly innervate cranial vasculature and dura mater, where the release of vasoactive inflammatory neuropeptides evokes head pain [69]. Evidence also suggests that before a migraine attack occurs, both endogenous pain modulation by descending pathways and the sensory threshold are reduced [36,72]. Indeed, women with MM had a lower nociceptive threshold during the hormone-free interval of CHC, when circulating estrogen levels are less measurable [73]. All three ER subtypes are highly expressed within the TG and their local activation can be associated with migraine pathology [36]. Indeed, females appear to have more ERs in TG than males [38]. In addition, both sex and treatment with estrogens influence nitroglycerin (NTG)-induced neuronal activation in the rat brain, a well-established experimental model to investigate nociceptive transmission, as well as neuroendocrine and autonomic functions [74]. ERα is localized in the nucleus or cytoplasm of numerous small and medium-sized trigeminal neurons [41], in glial cells, and in the nodes of Ranvier of thicker myelinated Aδ fibres [38]; even cytoplasmatic ERβ and membrane or cytoplasmatic GPER are found in most trigeminal neurons [38]. Estrogens also exert non-genomic acute effects on vascular tone through an antagonistic action on the calcium channels of smooth muscle cells [75] and an enhanced nitric oxide synthesis in endothelial cells [76], thus promoting vasodilation [38]. Furthermore, an estrogen-mediated mechanism contributes to cortical spreading depression (CSD) [77], a characteristic feature of migraine with aura consisting of a strong wave of neuronal depolarization with glial and vascular activation [78]. In experimental animal studies, CSD was shown to activate trigeminal nociceptors and facilitate a substantial inflammatory response relevant to head pain [79,80]. Interestingly, high E2 levels increase CSD susceptibility, while estrogen withdrawal decreases CSD [81]. These findings may explain why perimenstrual migraine attacks are less likely to be associated with aura in women with MRM [82].

### 2.5. Estrogens, Calcitonin Gene-Related Peptide, and Neuroinflammation

The calcitonin gene-related peptide (CGRP) is a sensory neuropeptide with a role in peripheral and central pain mechanisms including those leading to migraine [83]. It has vasoactive properties leading to meningeal irritation and neurogenic inflammation. Indeed, when released by axons of primary afferent nociceptive neurons of trigeminal nerves, CGRP induces plasma extravasation, vasodilation, and the degranulation of meningeal mast cells [84,85,86]. Circulating CGRP levels are elevated in migraineurs [87] and reproductive hormones influence the release of CGRP [36]. Indeed, blood levels of CGRP are higher in females than in males [88], increase throughout pregnancy [89] and under CHC use [86], and vary after menopause [90]. ERs are expressed in the same brain areas, including the TG, as CGRP receptors, thus indicating that estrogens can be involved in the modulation of CGRP signaling in the pathophysiology of migraine [36]. CGRP is produced in the small trigeminal neurons associated with nociceptive C fibers [91,92]. It is likely estrogens regulate these cells, via both genomic mechanisms and second messenger cascades [93]. Even the medium-sized trigeminal neurons associated with Aδ fibers co-localize the CGRP receptor and ERs, suggesting a modulation by both molecules [36]. In TG, ERα is found in neurons that co-express CGRP, and ERβ is colocalized with CGRP in the Golgi apparatus, whereas most CGRP positive cells express GPER [38]. However, the data concerning the association between estrogens and CGRP are controversial. Circulating CGRP concentrations were higher in low estrogenic conditions as in postmenopausal women [94] and in ovariectomized female rats which manifested both enhanced CGRP levels in the dorsolateral periaqueductal grey [95] and increased CGRP gene expression in TG [89]. The administration of exogenous estrogens in ovariectomized rats resulted in reduced CGRP levels [96], as well as in increased CGRP content in the dorsal root ganglion [97]. Another study revealed no changes in basal CGRP release from the peripheral terminals of meningeal nerves in both female and male rats [86]. Some neurons containing CGRP or CGRP receptors also express OT receptors, thus suggesting a possible dual neuronal modulation, either a pro-migraine effect through CGRP or an anti-migraine effect through OT [65].

Substance P (SP) is another vasoactive neuropeptide released from trigeminal nociceptive nerve fibers on cranial meninges, thus contributing to the neurogenic inflammation process underlying migraine pain [86]. Estrogenic supplementation decreased plasmatic SP levels in ovariectomized rats [98]. Moreover, Cetinkaya et al [86] observed that estrogens decreased SP release from TG without altering SP release from meningeal trigeminal nerve terminals. This evidence suggested that estrogens might exhibit different effects on SP release depending on the nervous structures involved [86].

In summary, CGRP and SP are biomarkers of neuroinflammation involved in the pathophysiology of migraine. Estrogens modulate CGRP and SP release through inhibitory effects; thus, they have a protective role against neurogenic inflammation [86]. Moreover, estrogens antagonize the inducing effect of progesterone on CGRP and SP release, suggesting that a combination of estrogens and progesterone may stabilize CGRP and SP release from pain-related structures [86].

Estrogens are also able to modulate the neuronal inflammatory response by interfering with nuclear factor kappa B (NF-κB) [18]. NF-κB is a transcription factor that mediates the expression of several inflammatory genes, including pro-inflammatory cytokines such as the tumor necrosis factor-alpha (TNF-α), interleukin-1β (IL-1β), interleukin-6 (IL-6), and other chemokines [98]. On one hand, it was observed that female migraineurs have increased levels of pro-inflammatory cytokines (TNF-α, IL-1β, IL-6, IL-8) during both the migraine attack and the migraine-free period [80]. On the other hand, in vitro experiments revealed that the activation of ERβ could dampen NF-κB signaling, thus reducing circulating inflammatory molecules [99]. Furthermore, estrogens seem to be protective against prostaglandins-induced inflammation because the estrogen-mediated lowering of the NF-κB pathway diminishes the production of cyclooxygenase, an enzyme involved in the synthesis of prostaglandins [98]. Conversely, estrogen withdrawal might increase the susceptibility to prostaglandins, which were able to facilitate neuroinflammation through the enhanced release of neuropeptides including CGRP, SP, and neurokinin [100].

That being so, it is likely that estrogens alone are not capable of exerting any direct anti-migraine effects, but they act throughout migraine circuitries influencing the balance of pro- and anti-migraine factors to increase the threshold and then to suppress the initiation of head pain, especially in the TG. In the presence of estrogens, a suppression of the trigeminal pathway occurs reducing migraine vulnerability, whereas in the absence of estrogens the opposite occurs with a lower threshold for migraine attacks [36].

## 3. Estrogens in the Treatment of Menstrual Migraine

The role of reproductive hormones is still under investigation but the gender-dependent prevalence of migraine underpins a strong attention to reproductive milestones and eventually to hormonal manipulations [6]. In some patients, migraines mainly occur during menstruation and have more severe clinical features than those experienced in other phases of the menstrual cycle [101,102].

In migraineurs, especially in those with migraine with aura, experts have discouraged hormonal treatments containing estrogens due to an increased risk of ischemic stroke [103,104,105] and disease worsening [106,107]. However, manipulations of natural estrogen fluctuations may represent a valuable option to prevent menstrual attacks [108]. Several studies have investigated the efficacy of hormonal treatments, which aim at reducing hormonal fluctuations and estrogen withdrawal preceding menstrual bleeding (Figure 1, top panel). Indeed, the “perimenstrual window” of vulnerability to migraine, specifically between −2 days and +3 days of menstruation, seems to be strongly related to hormonal withdrawal [109].

There are two main categories of hormonal treatments evaluated in MM prevention (Figure 1, middle panel). On one hand, CHC (oral, vaginal ring) to avoid unintended pregnancies with different regimens of ethynilestradiol (EE) administration according to the duration of the hormone-free interval (HFI) (standard regimens 21/7, variable extended regimens, continuous regimens without HFI) (Figure 1, bottom panel). On the other hand, estrogen treatments not ensuring contraception, which replace the fall in estrogen during the perimenstrual period both in patients with MRM and in those with PMM (Figure 1, middle panel).

Overall, data on hormonal treatments in MM prevention are scarce and heterogeneous but they suggest their ease of use, versatility, and good safety profile [105,108]. Estrogens have been administered as short-term preventive strategies during the perimenstrual period to replace the fall in endogenous estrogens in those suffering from PMM with predictable menstruations. They have been also administered at low doses in a continuous modality to counterbalance physiological hormonal fluctuations, or in combination with progestogens to stabilize hormonal levels and ensure contraception [105,108]. In the following paragraphs, we will discuss the available data on exogenous estrogen administration in women with PMM or MRM according to treatment formulations and administration routes.

### 3.1. Combined Hormonal Contraception (CHC)

CHC composition has continuously evolved over the years with the aim of improving safety, efficacy, tolerability, and non-contraceptive benefits. While the estrogen component has remained predominantly EE, due to its good bioavailability, formulations have varied significantly in terms of progestogens [110,111,112]. Concerns over dose-dependent health risks (e.g., venous thromboembolism [VTE] and cardiovascular [CV] events), mainly associated with EE, have led to the development of alternative formulations with either reduced doses of EE, natural estrogens, less androgenic progestogens, or different routes of administration [113].

Combined oral contraceptives (COCs) are the most popular type of CHC, with a wide range of hormones, doses, and regimes. The estrogenic component is mainly represented by EE, usually in a dosage of 20–30 μg, but the newest formulations today also deliver natural forms of estrogen, such as E2, estradiol valerate (E2V), the ester of E2, and estetrol (E4), a fetal form of estrogen [114]. The progestogenic component ensures the efficacy of the contraceptive and includes several types of progestogens at different doses depending on their biological potency [115]. The most common way of administrating COCs is the 21/7 regimen (i.e., 7 days of HFI) [116]. However, the availability of regimens with a shorter (6, 4, or 2 days) or absent HFI enables the prevention of the migraine worsening that sometimes characterizes the HFI. Indeed, observational studies showed that the reduction [117,118] or the absence [119] of HFI led to a significant decrease in the frequency, intensity, and duration of migraine attacks. Another strategy to improve MM during COCs use envisages estrogenic supplementation during the HFI to prevent estrogen withdrawal [120]. Both therapeutic approaches suggest that inhibiting the fall in estrogen or its replacement may be effective in the prevention of MRM and PMM (Table 1).

### 3.2. Short HFI

A significant amount of evidence has demonstrated the positive impact of a shorter HFI on MM (Table 1). A prospective randomized study compared the administration of a COC containing 20 μg EE and 3 mg drospirenone (DRSP) in a standard 21/7 versus a 24/4 regimen in women suffering from PMM [117]. Patients receiving the 24/4 regimen reported a significantly lower intensity and duration of migraine attacks occurring during menstruation since the first cycle, and results gradually improved throughout the study period [117]. Another prospective study evaluated the impact of a 26/2 regimen with a natural estrogen step-down (E2V from 3 mg down to 1 mg) and a progestin step-up approach (dienogest [DNG] from 2 mg to 3 mg) in women with MRM, both switchers from a standard 21/7 COCs and never users [118]. Overall, women reported a significant reduction in headache frequency and duration, and acute medication consumption (*p* < 0.001). Results were similar between those naïve to COCs and previous COC users. Researchers did not compare 21/7 with the 26/2 regimen [118].

### 3.3. Estrogen Supplementation during the HFI

A pilot study evaluated the impact on headache of a 21-day treatment with COC containing 20 μg EE followed by a 7-day treatment with 0.9 mg of oral conjugated equine estrogens in women with MRM [119]. All patients reported a 50% reduction in the number of headache days per cycle, and a remarkable reduction in headache severity with estrogenic supplementation in the HFI [119]. Transdermal patches represent another route to replace the fall in estrogen during COCs HFI: a crossover randomized placebo-controlled trial showed that 50 μg E2 patches tended to reduce the frequency and severity of headaches and the occurrence of some bothersome migraine-related symptoms such as nausea, but the results were not statistically significant when compared with the placebo [120].

### 3.4. Extended and Continuous Regimens

Two further studies evaluated the extended and continuous administration of COCs for 168 days both in patients with MRM [121] and in patients without a diagnosis of migraine but reporting headaches during a 21/7 COCs regimen [122]. The first study compared patients who previously received a 21/7 COCs with naïve patients. The study population all underwent a 168-day COC with 20 μg EE and 150 μg levonorgestrel [LNG]. A 4-day hormone-free interval was scheduled after 84 days of continuous treatment (extended regimen) or in the case of reported bleeding (flexible regimen). Overall, the hormonal treatment led to a lower headache severity and disability (measured using headache scores) in patients with MRM, and these results did not differ between patients previously receiving COCs in a 21/7 regimen and naïve patients [121]. The second study prospectively compared the evolution of the headache in a sample of switchers from a 21/7 regimen to a 168-day regimen with 30 μg EE DRSP 3 mg DRSP. The study population used CHC for purposes different from migraine prevention and had not received a diagnosis of migraine or MM according to the ICHD-3 criteria. Headache severity significantly reduced during the 168-day COC, and specifically in those who had previously reported more severe headaches during the 21/7 administration. The study had several limitations such as the gynecological setting and the evaluation of the headache mainly as a symptom of premenstrual syndrome, but the results were consistent with the aforementioned evidence on the efficacy of treatments reducing the HFI in MM prevention [122].

### 3.5. Contraceptive Vaginal Ring

The transvaginal ring is an alternative route of administration of CHC. It releases 15 μg/day of EE and 120 μg/day of etonogestrel (ENG) and its typical use follows a 21/7 regimen, with 3 weeks of vaginal delivery plus 1 ring-free week [123]. The vaginal route gives the advantage of stable hormonal levels during the 21-days of insertion [123]. In a randomized study, contraceptive ring users had EE serum concentrations 3.4 times lower than contraceptive patch users (EE 20 μg/norelgestromin [NGS] 150 μg) and 2.1 times lower than COC users (30 μg EE/150 μg LNG) [124]. No data are available on contraceptive patches and MM, whereas one study evaluated the efficacy of transvaginal hormonal treatment administered in an extended regimen in preventing MRM with aura [125]. A significant reduction in aura frequency and an MRM resolution was observed in 91.3% of patients, consistent with the stable hormonal levels and the absence of fluctuations allowed by the extended regimen [124]. The study did not provide safety data, but the authors speculated that a decreased aura frequency might positively affect the risk of stroke [125] as directly related to the frequency of auras [126].

### 3.6. Other Treatments with Estrogens

Table 2 summarizes the studies in which E2 was tested as a preventive treatment in women with MM. The percutaneous (patches or gel) use of E2, the main biologically active estrogen, represents a suitable alternative to the oral route of estrogen administration and offers the possibility to obtain more stable circulating levels because of the peculiar pharmacological properties [127]. Indeed, the transdermal route is not affected by gastric absorption and hepatic catabolism. Moreover, transdermal E2 has less impact on the synthesis of hepatic proteins, such as sex hormone-binding globulin (SHBG), coagulation factors, and angiotensinogen [127]. These findings can ensure a superior safety profile [128]. That being so, percutaneous estrogens should be preferred in women who suffer the most from hormonal fluctuations as it occurs premenstrually and in perimenopausal women [129]. The therapeutic window, namely treatment initiation and duration, varies across the studies; thus, the results should be cautiously interpreted and compared.

### 3.7. Transdermal Patches

Some studies evaluated the efficacy of transdermal E2 patches in MM prevention as a short-term estrogen therapy during the perimenstrual period (Table 2). In two randomized, placebo (PL)-controlled studies, the daily perimenstrual application of transdermal patches showed comparable results to PL in the prevention of PMM and MRM [130,131]. These two studies evaluated different doses and treatment schemes with similar results. The crossover trial evaluated E2 hemihydrate 50 μg patches applied from day −2 to day +4 of menstruation with no difference from PL in the frequency and intensity of headaches, and acute medication consumption [130]. The parallel PL-controlled study showed that 100 μg E2 patches applied from day −7 to day +7 were not superior to PL in reducing the number and intensity of menstrual attacks in patients with PMM [131]. However, their study design and their small sample size might have affected the results; moreover, methods to predict ovulation and the subsequent treatment start were less accurate than those now available, which could ensure a better definition of the therapeutic window. Another study compared three short-term preventive strategies for MRM with 25 μg E2 patches, 2.5 mg frovatriptan, or 500 mg naproxen sodium administered from −2 to day +4 of menses. Among the three, only frovatriptan significantly reduced headache frequency and severity [132].

### 3.8. Add-Back Therapy

Headache evolution has been studied in women using gonadotropin-releasing hormone (GnRH) analogue, an inducer of temporary iatrogenic menopause, plus transdermal E2 patches, alone or combined with progestogen (add-back therapy) (Table 2). In one case, researchers added 100 μg/daily transdermal E2 and 2.5 mg/daily oral medroxyprogesterone acetate (MPA) to 3.75 mg monthly GnRH administered for 10 months: patients reported a reduction in headache severity (*p* < 0.001) in both treatment phases (during GnRH administration alone and in combination with add-back therapy) [133]. In the other case, researchers compared transdermal patches of 100 μg E2 with PL administered 1 month after the implant of a GnRH releasing device: transdermal E2 proved superior to PL in reducing headache disability (*p* = 0.035), and severity (*p* = 0.03), but had no effects on headache frequency (*p* = 0.7) [134]. The study is not reported in Table 2 because patients with a general diagnosis of migraine were recruited and the reduction in the severity of perimenstrual headaches minimally influenced the overall efficacy outcome [134]. Of note, patients without a diagnosis of MM reported a beneficial effect of the add-back therapy, supporting the claim that the lack of hormonal fluctuations may also be useful to prevent migraine attacks occurring outside the perimenstrual period.

### 3.9. Transdermal Gel

A few trials have proved the superiority of 1.5 mg E2 gel to PL in short-term MM prevention. E2 was administered for 7 or 8 days (from day −2 to +5 [135,136] or from day −6 to +2 of the menstrual cycle [137]) in 2.5 g of gel. These studies showed a significant reduction in MM frequency (*p* < 0.01, *p* = 0.04, respectively) [135,137] with a great effect on moderate to severe attacks (*p* < 0.05; *p* = 0.003, respectively) [136,137], and a reduction in medication consumption [136]. A disease rebound might happen shortly after treatment discontinuation. Indeed, MacGregor et al. [137] reported a significant increase in migraine occurrence in the 5 days immediately following E2 use compared to PL (RR 1.40, *p* = 0.03), with a spontaneous resolution after 5 further days. This effect might be explained by an insufficient dose of hormone, or an inadequate start and treatment duration. Indeed, the first two randomized controlled trials started the short-term prevention later in the menstrual cycle and thus, they might have ensured longer E2 supplementation across menstrual bleeding [135,136]. Conversely, MacGregor et al. [137] started treatment 6 days before the expected menstruation to achieve the nadir of systemic estrogen concentration 2 days before menses, when migraines are more likely to occur. A limitation to this therapeutic strategy is represented by the possibility of having irregular bleeding with an unpredictable luteal phase. Further studies might provide information on the optimal treatment dose, duration, and start and might consider patients with migraine with aura.

### 3.10. Estradiol Implants

The sole study evaluating the efficacy of subcutaneous E2 implants in a few patients with MM proved the efficacy of this route of administration [138]. Patients received implants containing 100 μg E2 at first and subsequent maintaining doses of 50 μg together with cyclic 7-day oral norethisterone acetate (NETA) at 5 mg leading to endometrial shedding. Most patients registered headache improvement: 46% of women reported complete headache freedom and 37.5% almost complete symptom relief [138]. The treatment ensured stable levels of estrogen, but no safety data are available on the long-term exposure to high-dose estrogens.

### 3.11. Phytoestrogens

Phytoestrogens—natural agents exerting estrogenic activity in target tissues— such as genistein and daidzein, have been deemed helpful in the relief of hot flashes, menopausal symptoms, and MM. They exert their action as selective estrogen receptor modulators (SERMs) because of their structure which is similar to E2. An open label study showed the efficacy of 56 mg genistein and 20 mg daidzein administered 10 days per month (from day −7 to +3 of menstruation) as MM prophylaxis in 10 women [139]. Patients observed an average reduction of 62% of days with headache from the baseline (*p* < 0.005) and a reduction in headache intensity (*p* < 0.005); half of them also reported the resolution of autonomic symptoms (nausea and/or vomiting) related to migraines [139]. A randomized controlled trial confirmed the efficacy of 24-week treatment with 60 mg soy isoflavones, 100 mg dong quai, and 50 mg black cohosh, which proved superior to PL in reducing the frequency of MRM attacks and headaches, triptans, and other acute medication consumption, and headache severity (*p* < 0.001) [140]. Therefore, phytoestrogens represent a suitable therapeutic option for women not willing to take or with contraindications for hormonal treatments, such as patients at a high risk for stroke [139].

### 3.12. Other Hormonal Treatments

As part of CHC and estrogen-only treatments, different types and doses of progestin-only treatments have been studied in women with migraine, without a specific focus on MM [141,142,143,144,145]. Interestingly, when they were tested, among other contraceptive options, in a small sample of women with MM, the achievement of amenorrhea was associated with a significant reduction in headaches [146].

It is important to underline that progestogens have a lower stroke risk than estrogens [147] and are effective in preventing migraine with aura [141,145], making them a preferable option for women with risk factors [105]. It is likely that the use of progestins at ovulation-inhibiting dosages decreases cortical excitability by maintaining low estrogen levels and avoiding estrogen withdrawal [103]. The sole study comparing 75 μg oral desogestrel with an extended-cycle regimen of 150 μg COCs (desogestrel [DSG] and 20 μg EE) showed that both treatments were effective in migraine prevention, but the progestin-only contraceptive was superior in reducing the days with acute medication consumption (*p* = 0.044) [142].

Researchers also evaluated the role of testosterone (T) in migraine prevention and found that subcutaneous T implants in patients with androgen-insufficiency symptoms reduced the severity of headaches [148]. The T mechanisms of action in migraine pathogenesis are unknown, but preclinical studies showed that they could modulate the central neurotransmitter pathways suppressing CSD [149]. Interestingly, women following ovarian removal, a well-established condition of androgen insufficiency [150], reported a significantly higher rate of migraine in comparison with women who entered natural menopause [151].

As far as HRT is concerned, migraine women entering menopausal transition have a worsening of their head pain, probably because of the erratic release of estrogens [152]. Therefore, in women who still have irregular menstrual cycles the use of HRT may be helpful to manage estrogen-withdrawal migraine along with vasomotor symptoms [153]. Using only the lowest doses of estrogens necessary to control menopausal symptomatology minimizes the risk of unwanted side effects, especially in migraine with aura [153]. The transdermal route of E2 administration [154] or, whenever possible, the use of continuous combining regimens of HRT that avoid scheduled bleeding [155] or tibolone, a selective tissue estrogenic activity regulator with mild androgenic properties [156], seemed the best option to manage migraine at menopause.

### 3.13. Clinical Guidance

As shown in Table 1 and Table 2, studies have provided low-quality evidence. Most of them were carried out in gynecological settings and included small samples of women who required contraception or hormonal treatments for medical reasons other than MM prevention. Overall, some types of hormonal treatments have proved effective in reducing the frequency and burden of migraines related to hormonal fluctuations. The expert consensus suggests that extended-cycle regimens and treatments with a short hormone-free interval are preferred; if a 21/7 regimen is chosen, supplemental E2 (as oral or transdermal formulations) might be added during the hormone-free interval for short-term MM prevention [108]. Short-term prevention could also represent a valid therapeutic approach for those who do not need contraception and have predictable menstrual bleeding and migraine [108]. In terms of formulations, transdermal E2 and transvaginal CHC administration provide more stable levels of hormones avoiding the first pass metabolic effect (Figure 2).

### 3.14. Safety

The main concern for CHC in migraine is their vascular risk and, specifically, stroke risk [157]. Researchers have investigated the complex epidemiological association between stroke and migraine in depth, agreeing on a different risk category related to the migraine subtype: migraine with aura represents a 2-fold increased stroke risk, while migraine without aura has a minor risk [105,158,159,160,161,162]. Considering the low stroke risk in young women, the absolute number of additional strokes related to migraine is low [151,155] but not negligible because of the potential fatal or high disabling outcome of the event. Evidence suggests that migraine is an independent risk factor for stroke, similar to pro-coagulative factors or antiplatelet discontinuation [163], and that the coexistence of multiple stroke co-factors or promotors, such as thrombophilia, patent foramen ovale, white matter hyperintensities as signs of chronic ischemic brain sufferance, arterial dissection, and smoking, is quite common in migraineurs [164,165,166,167].

Estrogens represent a known risk factor for vascular events as they have a dual action in the vascular system. They improve endothelial-dependent blood flow and lipid profiles, but they also have pro-thrombotic and proinflammatory effects, which might be predominant in those with comorbidities or co-factors increasing the risk of stroke and other vascular events (i.e., myocardial infarction or deep venous thrombosis) [157,168]. COC is associated with an earlier stroke onset, an undetermined cause of stroke, less intra- and extracranial arterial stenosis, and a lower stroke recurrence, which might be explained by hormonal treatment discontinuation after the event [169]. The combination of two independent stroke risk factors—migraine and hormonal treatments—represents a concern; several studies have proved that female migraineurs receiving CHC have an increased stroke risk with an odds ratio ranging from 2.1 to 13.9 [166,167,170,171,172,173,174,175,176]. Stroke risk is estrogen dose-dependent: a case study showed that the risk halved in those receiving 20 μg EE and was even lower in those treated with progestin-only methods [171]. Although EE doses >50 μg have been abandoned because of the relevant vascular risk, low doses also confer an increased stroke risk, which is hard to estimate because of the heterogeneity of the available studies [163]. The retrospective design of most studies and the lack of information on the type of hormonal treatment employed require a cautious interpretation of the results. Future longitudinal studies evaluating the stroke risk in migraineurs taking modern hormonal treatments might address some of the existing knowledge gaps.

The European Headache Federation and the European Society of Contraception and Reproductive Health have recommended avoiding CHC in patients suffering from migraine with aura unless it is mandatory for comorbidities such as polycystic ovary syndrome; in these cases, gynecologists must individually choose the type of contraceptive [105]. Patients with migraine without aura should avoid CHC when other vascular risk factors coexist; otherwise, progestin-only contraceptives or COCs with doses of estrogens lower than 35 μg are preferable [105].

Data from studies evaluating the efficacy of hormonal treatments in MM prevention suggest that extended-cycle regimens, formulations ensuring stable levels of hormones (i.e., transvaginal administration or transdermal estradiol supplementation), and low doses of estrogens represent a suitable option for patients requiring hormonal treatments. Stroke or myocardial infarction were not among the adverse events of these studies, whose sample size, treatment duration, and follow-up were still limited. Even if progestin-only treatments have the safest vascular profile, unexpected breakthrough bleedings might affect treatment tolerability [141,143] (Figure 2). Future long-term data will be helpful in evaluating the feasibility of hormonal treatments in MM prevention, providing details on the preferred regimen.

## 4. Controversial Issues and Future Research Areas

Though temporally restricted to a limited portion of the month, MM represents an extremely disabling condition. Indeed, attacks of MM have been found to be more severe, disabling, and refractory to abortive medications than those that are non-menstrual related [177].

Migraine has a huge burden on the population below 50 years of age [178] and almost one-quarter of female patients with migraine have MM [179]. Altogether, these facts point to MM as an important neurological disorder that affects a relevant portion of the female population and needs adequate recognition [11].

### 4.1. Menstrual Migraine Definition

In this context, one main unsolved issue in need of priority attention is the limited visibility provided by ICHD-3 to MM. ICHD-3 does not list MM in the main body of the classification. In this large, detailed, and comprehensive nosographic apparatus, which has greatly contributed to making migraine one of the most active fields of scientific advance in the neurological area, MM is relegated to the Appendix among the disorders for which there is “uncertainty over whether they should be regarded as separate entities” [12] (ICHD-3).

This situation has historical and scientific reasons. In 1996 Anne MacGregor in her dissertation on “Menstrual Migraine: Towards a Definition” admitted that the term “menstrual” migraine is misused by both patients and doctors and lacks a precise definition [180]. She proposed a definition whereby the term “menstrual” migraine should be restricted to attacks exclusively starting on or between day 1 ± 2 days of the menstrual cycle in women who are free from attacks at all other times of the cycle. In the conclusions, she called for future studies that are necessary to support or refute the proposed definition. More than a quarter of century later, the lack of precise diagnostic criteria has led to conflicting results in studies on prevalence figures, clinical characteristics, and responses to treatment. Moreover, it is also likely to have reduced or diverted research on the efficacy of drugs specifically for MM. Importantly, clinical trials dealing with non-hormonal treatments often do not distinguish between perimenstrual attacks in women diagnosed with MM and attacks associated with menstruation by chance or, in some cases, pool subpopulations with PMM and MRM, without specifically analyzing menstrual attacks [181].

Attempts have been made over time to define the optimal time period for MM definition. From the brilliant original description by Somerville [14], which was, however, limited to a very small number of subjects, several surveys have been conducted to investigate menstrual-related migraines. Unfortunately, the level of evidence they yielded is low due to possible reporting biases. Mattsson surveyed more than 700 women and concluded that 21% of women with migraine without aura and 4% of women with migraine with aura reported that at least 75% of their attacks occurred within –2 to +3 days of menstruation [182]. The ages ranged from 40 to 74 thus warranting the idea that a relevant portion of subjects may have incurred reporting and recall bias, particularly postmenopausal women asked about their prior menstrual symptoms. Stewart et al. [183] surveyed 81 menstruating women between the ages of 18 and 55 that self-recorded their migraines and menstrual cycles over 98 days. They reported a significant increase in the risk of migraine attacks from day −2 to day 1 [183]. The 28 women aged between 22 and 29 surveyed by Johannes et al. [184] via 4-month self-reported diaries showed a higher risk of migraine during the first 3 days of menstruation (odds ratio, 1.66; 95% confidence interval, 1.21 to 2.26), but headache risk was not significantly increased during the 2 days immediately preceding the onset of menstruation.

There is the possibility that PMM and MRM reflect two different situations. In MRM, the occurrence of migraine attacks outside of menstruation may suggest that the fall in estrogen is just one of the triggers of migraine attacks. Conversely, in PMM the relation between migraine and menstruation is exclusive and may therefore imply a specific pathophysiological mechanism. Future research should focus on PMM, verifying whether it actually exists, its epidemiological impact, and its temporal pattern. Hopefully, clarification of these aspects will pave the way for the identification of biomarkers specific to this subtype of migraine and possibly to tailored treatments. In this context it is worth noting that a human model exists for MM which is represented by a migraine attack occurring during the bleeding associated with HFI in women on 21/7 hormonal contraception. The model does not replicate in full the condition of ovulating women, as exogenous hormones suppress the hypothalamic–pituitary–ovarian axis, but it has the advantage of precisely predicting the occurrence of bleeding and attacks. Using such a model, we were able to report a reduction in the pain threshold in a population of migraine patients whose attacks were associated with the HFI [73]. Further studies also involving MRM and women with migraine not related to menstruation are likely to provide important inputs for advancing knowledge on PMM.

### 4.2. Use of Headache Diaries

Headache diaries and calendars are extremely useful in several steps of headache diagnosis and management [185,186,187]. They are even more useful for the diagnosis of MM, where the prospective use of a headache and menstruation diary is required to verify the three main characteristics of MM. They include: (1) type of migraine (without aura or with aura); (2) timing of attacks in relation to menstruation; and (3) frequency of attacks in relation to menstruation across different cycles (at least two out of three consecutive menstruations according to the Appendix criteria of ICHD-3). The use of a prospective diary is a fundamental step in gathering precise information because studies indicate that women tend to over-report an association between migraine and menstruation [188,189].

In the digital era we are living in, the adoption of a diary is facilitated by the ample availability of freely usable apps, some of which have been developed according to expert advice and provide user-friendly interfaces for patients associated with the possibility of downloading or sharing summary reports or granular data with their physicians [190].

In the research field, the use of headache diaries in association with sophisticated statistical methods will likely provide sufficient material to identify the temporal window for the definition of menstrual attacks and for separating those attacks that are part of MM from those that casual recur during subsequent menstruation. Experiences in this direction have already been undertaken and have shown the limitations of the ICHD-3 criterion based on the recurrence of menstrual attacks in two out of three menstruations. Barra et al. [191] have indeed demonstrated the superiority of a mathematical method to model MM attacks over the less specific two-out-of-three criterion.

### 4.3. Pathophysiological Insights

As discussed in the initial part of this review, there is a close relationship between the fall in pre-menstrual estrogen and the risk of migraine attacks. It is, however, worth noting that estrogens also fall during the follicular phase, but evidence suggesting the occurrence of a “follicular” migraine attack is at best limited [109] as are the attempts to search for alternative explanations for the fall in estrogen. Menstruations are also associated with a fall in progesterone, but results in this context are scarce and conflicting [192,193]. In addition, scattered reports suggest an alternative, non-hormonal explanation for MM attacks. Calhoun and Gill [194] showed lower ferritin levels in the plasma of women with recurrent migraine attacks during or toward the end of their menstruations. In agreement, a controlled study conducted in a large population of women suffering from migraine showed that iron-deficient anemia was associated with PMM and MRM [195].

### 4.4. Future Studies on Efficacy and Safety

As illustrated in the central body of this review, several studies have been conducted to evaluate the efficacy and safety of estrogens in MM. Unfortunately, they have provided low-quality evidence, mostly due to methodological limitations and to the small populations investigated. Collectively, the results obtained favor the efficacy of estrogens, which suggests the opportunity of further evaluating the performance of estrogens in properly designed and adequately powered randomized and controlled clinical trials.

A fundamental pre-requisite is of course the careful characterization of the population enrolled based on precise diagnostic criteria. On the other hand, it will be of paramount importance to design future research on MM focusing on the impact of new CHCs containing natural estrogens. Available data have already revealed the benefits of the E2V/DNG formulation on MM without aura [118]. The shorter HFI is certainly protective against the estrogen withdrawal that triggers migraine, but the molecular characteristic of natural estrogens along with the type of progestogens could also play a role in limiting the metabolic impact [196,197]. Indeed, a favorable cardiovascular and metabolic profile of E2V, combined with DNG [198], and E2 with nomegestrol acetate (NOMAC) [199] has been shown, thus making natural estrogens ideal candidates to alleviate MM. Even the brand-new COC-containing E4/DRSP administered in a 24/4 regimen has had encouraging results on the hemostatic parameters [200].

Therefore, we speculate that CHCs with natural estrogens could become the gold standard for women suffering from MM, both for their possible favorable impact on migraine and for their lower cardiovascular and thromboembolic risks. Further studies are warranted in this field, and research should also address women with migraine with aura.

## Figures and Tables

**Figure 1 cells-11-01355-f001:**
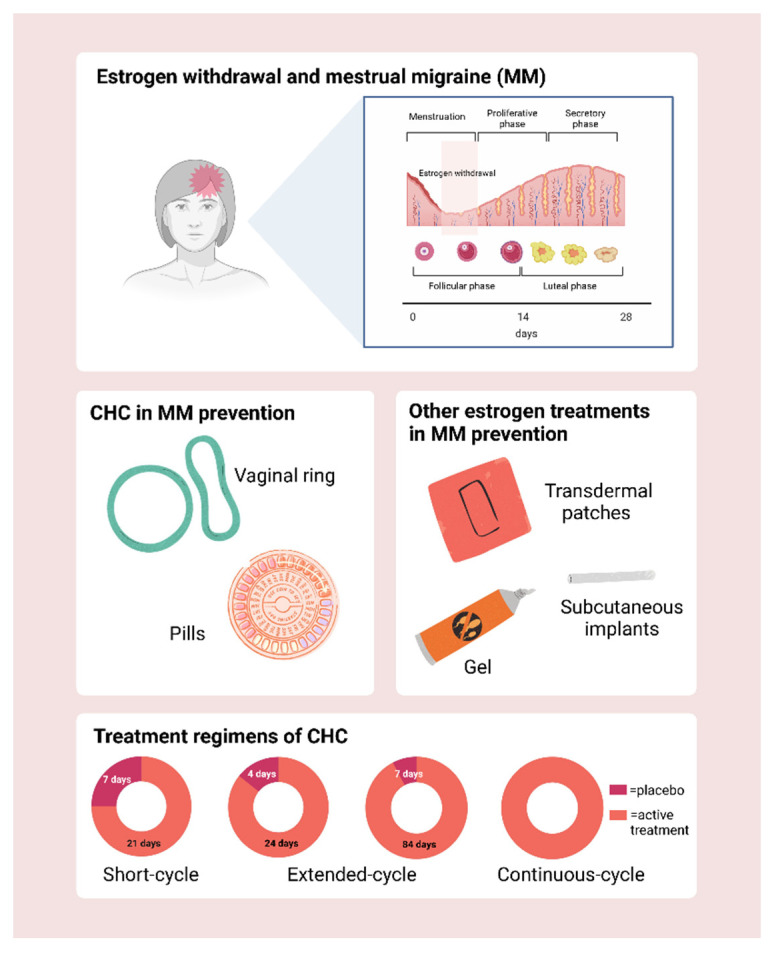
Schematic phases of the menstrual cycle (top panel); types of combined hormonal contraception (CHC) and other estrogen treatments evaluated in menstrual migraine (MM) prevention (middle panel); main treatment regimens of CHC (bottom panel). Created with BioRender.com (2022).

**Figure 2 cells-11-01355-f002:**
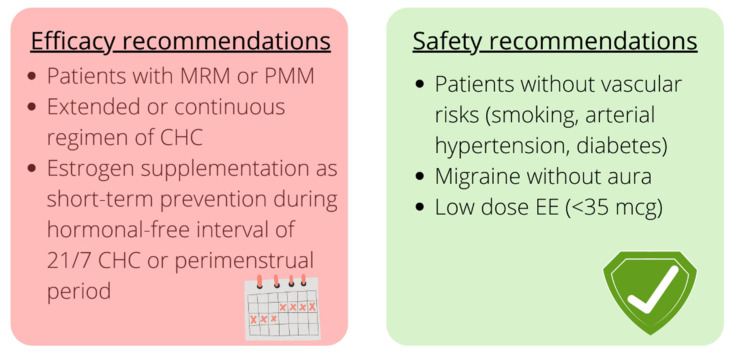
Summary of the main efficacy and safety recommendations for the use of hormonal treatments in menstrual migraine (MM) prevention. Created with BioRender.com (2022). Abbreviations: CHC: combined hormonal contraception, EE: ethinylestradiol, MRM: menstrually related migraine, PMM: pure menstrual migraine.

**Table 1 cells-11-01355-t001:** Main characteristics of the studies on Combined Hormonal Contraception (CHC) in the treatment of MM.

Study Year	Type of Study	Type of Migraine	Type of Treatment	Treatment Regimen	Treatment Duration, N Menstrual Cycles/Months	Sample Size	Age, Years ± SD or Range	Efficacy (Yes/No)
** * Oral * **								
De Leo et al., 2011 [117]	RCT	PMM	EE 20 μg + DRSP 3 mg	21/7 vs. 24/4	3	60	28.15 ± 6.96 27.77 ± 7.44	Yes (24/4 superior to 21/7)
Nappi et al., 2013 [118]	open label	MRM	step-down E2V + step-up DNG	26/2	6	28	40.6 ± 3.5	Yes
Calhoun et al., 2004 [119]	open label	MRM	COCs with EE 20 μg + 0.9 mg CEE	21/7 COCs + short prevention for 7 days (22–28)	2	11	41 (28–50)	Yes
MacGregor et al., 2002 [120]	crossover PL-controlled trial	migraine during the HFI	COCs with different doses of EE + 50 μg E2 patches	21/7 COCs + short prevention for 7 days (22–28)	4	13	33 (24–42)	No
Coffee et al., 2014 [121]	RCT	MRM	EE 30 μg + LNG 150 μg	21/7 COCs vs. 168 extended-cycle regimen	6	32 (21 no prior COCs users; 11 prior COCs users)	33.5 ± 6.8 no prior COCs users; 33.9 ± 6.7 prior COCs users	Yes
** * Transvaginal ring * **								
Calhoun et al., 2012 [125]	open label	MRM with aura	EE 15 μg + ENG 0.120 mg + E2 75 μg transdermal patches	84/7 + short prevention for 7 days (from 85 to 91)	3	23	32.4 (19–55)	Yes

Abbreviations: CEE: conjugated equine estrogens; COCs: combined oral contraception; DNG: dienogest; DRSP: drospirenone; ENG: etonorgestrel; E2: estradiol; E2V: estradiol valerate; EE: ethinylestradiol; HFI: hormone-free interval; LNG: levonorgestrel; MM: menstrual migraine; MRM: menstrually related migraine; NETA: norethisterone acetate; N: number; PL: placebo; PMM: pure menstrual migraine; RCT: randomized controlled trial; SD: standard deviation.

**Table 2 cells-11-01355-t002:** Main characteristics of the studies on estrogen use in the treatment of MM.

Study Year	Type of Study	Type of Migraine	Type of Treatment	Treatment Regimen	Treatment Duration, N of Menstrual Cycles/Months	Sample Size	Age, Years ± SD or Range	Efficacy (Yes/No)
** *Estradiol patches* **								
Smite et al., 1994 [130]	PL cross over trial	PMM	E2 50 μg transdermal patches	short-term (from −2 to day +5)	3	20	(30–48)	No
Almen-Christensson et al., 2011 [131]	PL controlled RCT	PMM	E2 100 μg transdermal patches	short-term (from −7 to day +7)	3	27	39.6 ± 4.3	No
Guidotti et al., 2007 [132]	open label	MM	E2 25 μg patches, frovatriptan 2.5 mg, naproxen sodium 500 mg	short-term (from −2 to day +4)	NA	38 (14 frovatriptan; 20 E2 patches, 14 naproxen sodium)	29 ± 4 (E2 patches arm)	No
** *Add-back therapies* **								
Murray et al., 1997 [133]	open label	PMM	GnRH-A 3.75 mg, E2 100 μg + MPA 2.5 mg	10 months with GnRH-A (4 alone +6 with hormonal treatment)	15	5	NA	Yes
** *Estradiol gel* **								
de Lignieres et al., 1986 [135]	crossover PL-controlled trial	PMM	E2 1.5 mg	short-term (from −2 to day +5)	3	20	42.5 (32–53)	Yes
Dennerstein et al., 1988 [136]	crossover PL-controlled trial	PMM	E2 1.5 mg	short-term (from −2 to day +5)	4	22	39.8 ± 3.95	Yes
MacGregor et al., 2006 [137]	crossover PL-controlled trial	PMM or MRM	E2 1.5 mg	short-term (from −6 to day +2)	6	35	43 (29–50)	Yes
** *Estradiol implants* **								
Magos et al., 1983 [138]	open label	PMM	E2 implants: 100 mg/75 mg/50 mg + NETA 5 mg	Continuous estrogens + 7 days progestogen	up to 5 years	24	40.6 (32–51)	Yes

Abbreviations: E2: estradiol; GnRH-A: gonadotropin releasing hormone agonist; N: number; MM: menstrual migraine; MPA: medroxyprogesterone acetate; MRM: menstrual-related migraine; PL: placebo; PMM: pure menstrual migraine; RCT randomized controlled trial.

## Data Availability

Not applicable.

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
