# Peer review of "Role of Estrogens in Menstrual Migraine"

_cells, 2022, doi:10.3390/cells11081355_

Round 1

Reviewer 1 Report

The present manuscript is an in-depth narrative review on the relevance of estrogens in the pathophysiology and treatment of menstrual migraine.  This is a very interesting topic both from research and practice standpoints.

The manuscript is well organized and fully referenced. However some minor comments are listed below to further improve the quality of this review.

1) even though the topic is clearly  estrogens, it could be of interest for the readers if the author would expand a bit more on the paragraph about progesterone/progestins and androgens discussing also why this subject seems overlooked in the literature.

2) the authors focused their attention on combined hormonal contraception. It could be of interest to add a short paragraph on artificial menstrually-associated migraine induced by hormonal replacement therapy at menopause. Indeed, even if, nowadays, less common in clinical practice, such a topic may be relevant to cardiovascular and stroke risk in women. 

Author Response

The present manuscript is an in-depth narrative review on the relevance of estrogens in the pathophysiology and treatment of menstrual migraine.  This is a very interesting topic both from research and practice standpoints.

The manuscript is well organized and fully referenced. However some minor comments are listed below to further improve the quality of this review.

We thank the reviewer for appreciation of our work and for the time spent to improve our manuscript.

1) even though the topic is clearly  estrogens, it could be of interest for the readers if the author would expand a bit more on the paragraph about progesterone/progestins and androgens discussing also why this subject seems overlooked in the literature.

Some sentences marked in yellow have been added on the topic.

2) the authors focused their attention on combined hormonal contraception. It could be of interest to add a short paragraph on artificial menstrually-associated migraine induced by hormonal replacement therapy at menopause. Indeed, even if, nowadays, less common in clinical practice, such a topic may be relevant to cardiovascular and stroke risk in women.

We have added a short paragraph on HRT marked in yellow under the subchapter on other hormonal treatments.

Reviewer 2 Report

In the present paper evidences supporting the role of estrogens in the pathophysiology of migraine were reported in details, evaluating the numerous estrogens interactions with other neurotransmitters and molecules involved in pain modulation in cerebral areas known to be interested in migraine.

In the second part of the review were analyzed the efficacy and safety of prescribing exogenous estrogens as potential treatment of menstrually related migraine, comparing different type of treatment (EE, E2V, phytoestrogens) different treatment regimen and different route of administration(oral, transdermal patch, subcutaneous implants, vaingal ring).

Finally, authors pointed to controversial issues and future research areas in the field of reproductive hormones and migraine, suggesting the need to focus on PMM to identify biomarkers specific, with the aim to found more efficient treatment and to provide further knowledge on PMM. Moreover authors debate the controversial efficacy of estrogens, suggesting future research on MM focusing on the impact of the use of natural estrogens.

Overall, this paper reported a comprehensive evaluation of the state of art of the role of estrogen in migraine, and stimulates further researches to better understand the mechanism of action of estrogens, and in particular natural estrogens, in the modulation of migraine symptoms. 

Author Response

In the present paper evidences supporting the role of estrogens in the pathophysiology of migraine were reported in details, evaluating the numerous estrogens interactions with other neurotransmitters and molecules involved in pain modulation in cerebral areas known to be interested in migraine.

In the second part of the review were analyzed the efficacy and safety of prescribing exogenous estrogens as potential treatment of menstrually related migraine, comparing different type of treatment (EE, E2V, phytoestrogens) different treatment regimen and different route of administration(oral, transdermal patch, subcutaneous implants, vaingal ring).

Finally, authors pointed to controversial issues and future research areas in the field of reproductive hormones and migraine, suggesting the need to focus on PMM to identify biomarkers specific, with the aim to found more efficient treatment and to provide further knowledge on PMM. Moreover authors debate the controversial efficacy of estrogens, suggesting future research on MM focusing on the impact of the use of natural estrogens.

Overall, this paper reported a comprehensive evaluation of the state of art of the role of estrogen in migraine, and stimulates further researches to better understand the mechanism of action of estrogens, and in particular natural estrogens, in the modulation of migraine symptoms.

Thank you very much for appreciation of our work and for the time you have spent in revising our manuscript.